

# Heterogeneous mission planning for a single unmanned aerial vehicle (UAV) with attention-based deep reinforcement learning

Minjae Jung and Hyondong Oh

Mechanical Engineering, Ulsan National Institute of Science and Technology, Ulsan, South Korea

## ABSTRACT

Large-scale and complex mission environments require unmanned aerial vehicles (UAVs) to deal with various types of missions while considering their operational and dynamic constraints. This article proposes a deep learning-based heterogeneous mission planning algorithm for a single UAV. We first formulate a heterogeneous mission planning problem as a vehicle routing problem (VRP). Then, we solve this by using an attention-based deep reinforcement learning approach. Attention-based neural networks are utilized as they have powerful computational efficiency in processing the sequence data for the VRP. For the input to the attention-based neural networks, the unified feature representation on heterogeneous missions is introduced, which encodes different types of missions into the same-sized vectors. In addition, a masking strategy is introduced to be able to consider the resource constraint (*e.g.*, flight time) of the UAV. Simulation results show that the proposed approach has significantly faster computation time than that of other baseline algorithms while maintaining a relatively good performance.

# INTRODUCTION

Recently, mission environments such as disaster management or logistics services become more complex and larger. The goal of these large-scale missions can be achieved safer and faster using unmanned aerial vehicles (UAVs) (*Shakhatreh et al., 2019*; *Grzybowski, Latos & Czyba, 2020*; *Kim et al., 2021*). Since the complexity of task allocation by scheduling a large number of tasks in the mission to UAVs is high, it takes a long time for the human operator to plan these tasks manually without ensuring the optimal performance. The performance and computational time significantly impact the success rate of rescue in disaster management or the benefits of companies in logistics services (*Atyabi, MahmoudZadeh & Nefti-Meziani, 2018*). Therefore, autonomous mission planning algorithms need to be developed to solve these problems rapidly and efficiently.

Mission planning problems of the UAV can be represented as one of vehicle routing problems (VRPs). The VRP has various variations such as distance constraint (*Karaoglan, Atalay & Kucukkoc, 2020*), multiple trip availability (*Paradiso et al., 2020*), and

Corresponding author
Hyondong Oh, h.oh@unist.ac.kr

asymmetricity of costs (*Ban & Nguyen, 2021*) among many others. Thus, the VRP can represent some of the mission planning problems for the UAV, which is the complex combinatorial optimization problem. In addition, there are various studies to solve these VRP variations in operation research or transportation research fields (*Kumar & Panneerselvam, 2012*). Therefore, when representing and solving the mission planning problem of the UAV as one of the VRPs while considering the characteristics of the UAV, it is possible to develop more realistic mission planning algorithms with the help of VRP studies.

The VRP is a combinatorial optimization problem expressed in a graph form and there are (i) exact solvers, (ii) heuristic algorithms, and (iii) machine learning-based approaches to solve the problem. The exact solver approach can obtain the optimal solution with the branch and bound algorithm (*Laporte & Nobert, 1983*; *Toth & Vigo, 2002*; *Larrain et al., 2019*) or dynamic programming (*Secomandi, 1998*; *Mingozzi, Roberti & Toth, 2013*). Although the exact solver approach can achieve the optimal cost, the computation time grows exponentially with the scale of the problem. The heuristic algorithm approach finds a feasible solution much faster than the exact approach. There are various heuristic algorithms to solve the VRP such as variable neighborhood search (*Bräysy, 2003*; *Kytöjoki et al., 2007*; *Hemmelmayr, Doerner & Hartl, 2009*), tabu search (*Gendreau, Hertz & Laporte, 1994*; *Fu, Eglese & Li, 2005*; *Qiu et al., 2018*), and genetic algorithm (*Baker & Ayechew, 2003*, *da Costa et al., 2018*, *Ruiz et al., 2019*). Although the heuristic algorithms provide reasonable performance, they need to be designed carefully for different problem setting, which is often challenging and requires expert knowledge. The machine learning-based approach utilizes data to train model parameters. If the model can approximate the solver which maps input and output of the combinatorial optimization problem, it can be trained flexibly with different setting of the problem given the sufficient amount of data (*Khalil et al., 2017*). Besides, after training the model, fast computation time can be achieved. Using the machine learning approach is a good option when data is available thanks to its advantages of fast and flexible calculations; hence, this article adopts machine learning approach for UAV mission planning problems.

Among the machine learning approaches, supervised learning and reinforcement learning can be considered to tackle the VRP. *Vinyals, Fortunato & Jaitly (2015)* proposed a neural network model, called the Pointer network, that approximates the solver of combinatorial optimization. It uses the attention mechanism and the recurrent neural network (RNN)-based encoder-decoder structure with supervised learning. However, supervised learning needs a large amount of labeled data obtained from exact solvers that require significant time for data generation. For this reason, the reinforcement learning approach that generates data while interacting with the environment during the training process is often preferred for the VRP (*Bello et al., 2017*; *Kool, van Hoof & Welling, 2019*; *Mazyavkina et al., 2021*). *Bello et al. (2017)* proposed a reinforcement learning algorithm to solve combinatorial optimization problems that uses the Pointer network structure from (*Vinyals, Fortunato & Jaitly, 2015*), and then optimizes the model with the policy gradient

algorithm. *Kool, van Hoof & Welling (2019)* utilized the Transformer (*Vaswani et al., 2017*) style-neural network model that modifies the RNN structure of the Pointer network into multi head attention (MHA) network. This model is proposed to solve various types of routing problems with its flexibility, and the authors show that the model outperforms some heuristic algorithms and Pointer network-based model in terms of the performance and the computation time; hence, we build our approach based upon this MHA network.

The proposed approach in this study particularly considers the characteristics of the UAV, which are the capability of handling heterogeneous missions and the flight time constraint. As UAV technology development continues, several tasks can be carried out by even a single UAV simultaneously or sequentially, such as delivering extra payloads, visiting specific areas to take images of landmarks, or flying over large areas to obtain information. Considering these heterogeneous tasks, the cost for completing each pair of tasks becomes different depending on the order of task completion. For instance, different path lengths for delivery or radius of the area for coverage make the cost matrix of the VRP asymmetric. Another characteristic of a UAV is that they have limited flight time due to fuel/battery capacity. This constraint makes a UAV refuels/recharges their fuel/battery at the depot and resume their work. Therefore, the mission planning problem of a UAV in this article is represented as the multi-trip asymmetric distance constrained VRP (MTAD-VRP) which is one of the variations of the VRP. The heterogeneous mission planning considering these characteristics further increases the complexity of the VRP.

It is worthwhile noting that there are a few studies on heterogeneous mission planning problems for the UAV using heuristic optimization algorithms. *Zhu et al. (2018)* formulated the heterogeneous mission planning problem for UAV reconnaissance as multiple Dubins travelling salesman problem (MDTSP), which is one of VRPs and proposes the genetic algorithm-based approach to solve the problem. *Chen, Nan & Yang (2019)* considered an additional constraint which is time window and formulates the heterogeneous mission planning problem as a multi-objective, multi-constraint nonlinear optimization problem. Then, they utilize the search-based algorithm for optimization. *Gao, Wu & Ai (2021)* proposes ant colony-based algorithm for minimizing the weighted sum of the total UAV fuel consumption and the task execution time. The performance of the proposed algorithm is compared with other ant colony-based algorithms through numerical simulations. However, to our best knowledge, it is difficult to find the heterogeneous mission planning based on reinforcement learning approaches. As mentioned earlier, reinforcement learning-based approaches are expected to provide the superior performance compared with heuristic algorithms in terms of computation time and optimality. Besides, aforementioned works consider only single-trip problems which significantly limit the capability of the UAV.

To this end, this study proposes an attention-based reinforcement learning algorithm for the heterogeneous mission planning for a single UAV. We first formulate the heterogeneous mission planning problem as the MTAD-VRP expressed in a graph form to utilize the solvers for the VRP. Considering a realistic complex mission environment and characteristics of the UAV, we use the reinforcement learning approach with an attention-based neural network model to solve the problem with its fast computation time and

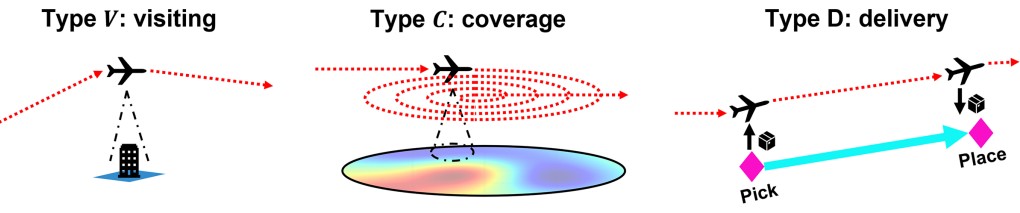

**Figure 1 Illustration of heterogeneous missions.**

flexibility. Although the existing learning-based algorithms can deal with various routing problems, most of them only consider homogeneous inputs (*Vinyals, Fortunato & Jaitly, 2015*; *Bello et al., 2017*; *Kool, van Hoof & Welling, 2019*). Thus, we introduce the unified mission representation for network inputs to contain the information of heterogeneous missions. And then, we design the masking strategy to deal with the flight time constraint to complete all tasks in a mission and help the training process of reinforcement learning. The proposed algorithm uses the MHA-based model architecture for better computational efficiency than that of the RNN-based model architecture while preventing the vanishing gradient effect when dealing with long data sequences. Furthermore, the MHA-based model has a permutation invariant property that makes the model to be able to learn the robust strategy regardless of input permutation. The REINFORCE algorithm (*Sutton et al., 2000*) with a baseline updates the model to converge stably by reducing the variance of the parameters' gradient. To validate the feasibility and the performance of the proposed approach, we perform numerical simulations and compare the result with state-of-the-art open-source heuristic algorithms.

## PROBLEM DEFINITION

This study considers visiting, coverage, and delivery as heterogeneous missions. Here, visiting is for capturing an image of the landmark building, coverage is for gathering information of a large area with the spiral flight pattern, and delivery is for picking and placing the package. Figure 1 illustrates heterogeneous missions. Note that, if needed, more mission types could be readily incorporated into the problem thanks to the flexibility of the learning approach.

Our purpose is to complete all the given heterogeneous mission while minimizing the flight time of a single UAV dispatched from the depot. The flight time budget constraint should be satisfied, and the UAV is allowed to return to the depot for recharging. Figure 2 shows a sample mission scenario in a 2-D view, where the black squares are the depot, blue squares are visiting mission spots, circles are coverage mission areas, and a pair of magenta diamonds with cyan arrows are the delivery mission with a specific direction.

To formulate the heterogeneous mission planning problem as the mathematical formulation of the VRP, we abstract the problem into a graph instance. The mission graph $G = (V, E)$ consists of $k$ nodes $(v_1, v_2, \cdots, v_k) \in V$ and edges $(e_{12}, e_{21}, \cdots, e_{k1}, e_{1k}) \in E$. Nodes represent the feature of each mission and the value of edges are constructed with the

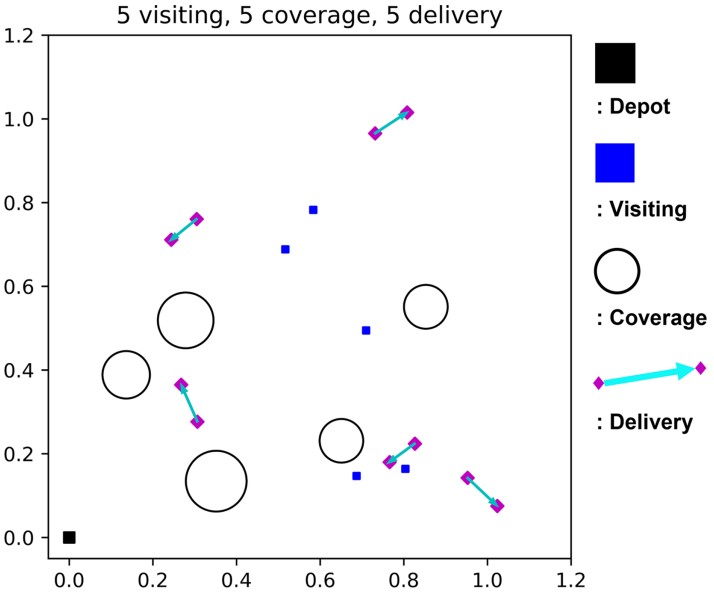

**Figure 2 A sample mission scenario with five visiting, five coverage, and five delivery missions.**

travel time cost which depends on the type of mission. The solution of the problem is the sequence of the index of nodes $\Omega = (n_1, \cdots, n_t)$, where $n_t \in \mathbf{N}$ and $1 \leq n_t \leq k$. The total travel time cost $L$, which is the objective of the problem is the sum of every value of edges between the selected nodes and the cost of returning to the depot as:

$$L = \sum_{t=1}^{k-1} e_{n_t n_{t+1}} + e_{n_k n_1} \quad n_t \in \Omega. \tag{1}$$

Assuming that the UAV flies with a constant velocity, the travel time cost between missions is calculated by the total distance that the UAV need to fly. We ignore the time of recharging and loading packages for simplicity. The type of mission affects the cost calculation as:

$$c_{xv} = d, \tag{2}$$
$$c_{xc} = d + S, \tag{3}$$
$$c_{xd} = d + l, \tag{4}$$

where $S = \pi r^2 / w$, $c_{xv}$, $c_{xc}$, and $c_{xd}$ are the travel cost to the visiting, coverage, and delivery mission point, respectively, from the source mission point $x$. $d$ is the distance between missions, $S$ is the length of the spiral path to cover the area, $w$ is the sensing range of the UAV, $r$ is the radius of the coverage area, and $l$ is the length of the delivery path. The cost for returning to the depot is the same as $c_{xv}$ with the visiting mission point of the depot. Figure 3 provides the conversion of the mission instance to the graph representation.

Additionally, we consider the limited flight time constraint of the UAV for safe mission completion. Typically, the UAV can be recharged at the base station which is considered as

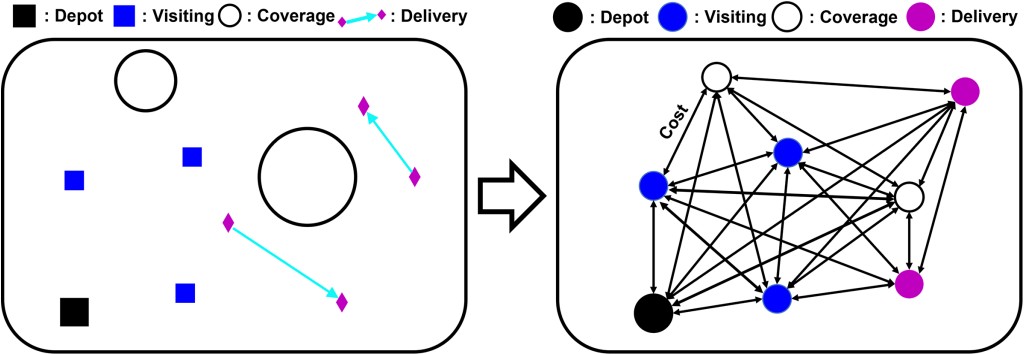

**Figure 3 Visualization of abstracting a mission instance as a graph instance.**

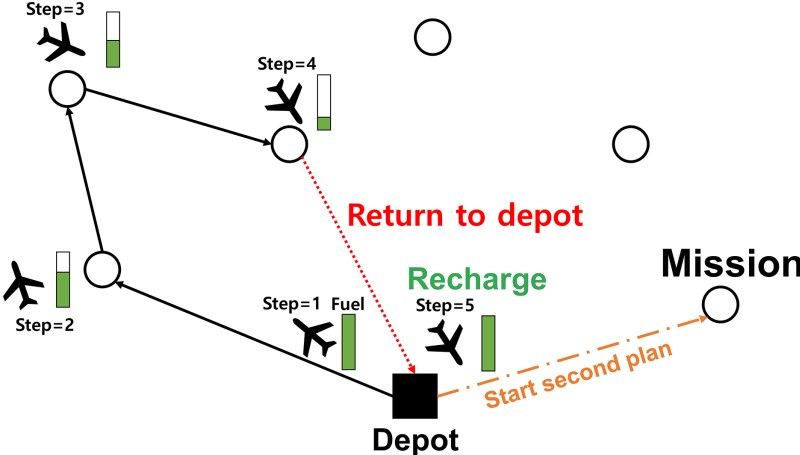

**Figure 4 Illustration of recharging the UAV.**

the depot in the VRP. Thus, we allow the UAV to be recharged by revisiting the depot. Figure 4 illustrates the recharging event graphically.

# ATTENTION BASED DEEP REINFORCEMENT LEARNING

In this section, we first propose a unified feature representation to deal with heterogeneous missions. Then, we suggest a masking strategy to consider the flight time limitation constraint. We introduce the neural network model and reinforcement learning algorithm to solve the heterogeneous mission planning problem using these methods. The neural network model architecture consists of an encoder and decoder network with the attention mechanism for sequential data. The REINFORCE algorithm (*Sutton et al., 2000*), one of the reinforcement learning algorithms, is used to optimize the neural network.

## Unified feature representation

We propose the unified feature representation $v = (x_1, y_1) || (x_2, y_2) || A || I_{Type}$ combining the spatial information of heterogeneous missions and indicator of each mission type, where $(x_1, y_1)$ is the critical position of the mission, $(x_2, y_2)$ is the end-position, $A$ is the

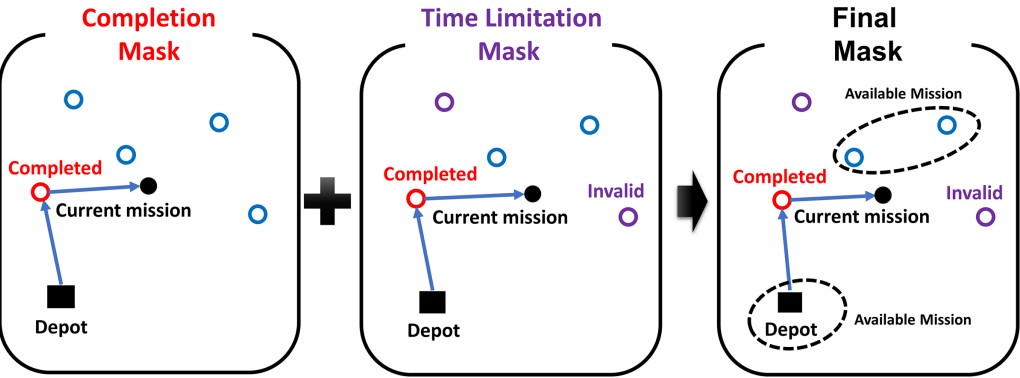

**Figure 5 The example of the masking strategy.**

area information, and $I_{Type}$ is the one-hot encoding indicator of each type. The operator ||
means concatenation. The critical position represents the important position of each
mission such as the position of the visiting mission, center position of the area of the
coverage mission, and picking position of the delivery mission. The end position
represents the position when the UAV completes the current mission. The delivery
mission has a different end position while the others have the same end position as the
critical position. The area information represents the radius of an area for the coverage
mission, the length of the delivery mission, or zero for the visiting mission. The depot has
the same representation as the visiting mission except $I_{Type}$. $I_{Type}$ represents the type
information of each mission, where $(0, 0, 0), (0, 0, 1), (0, 1, 0)$ and $(1, 0, 0)$ represent the
depot, the visiting mission, the coverage mission, and the delivery mission, respectively.

## Masking strategy

The masking strategy generates the mask $M$ to prevent selecting invalid actions in
reinforcement learning. The mask consists of the completion mask $M_C = (m_{c_1}, m_{c_2},
\cdots, m_{c_k})$ for already completed missions and the time limitation mask $M_T = (m_{t_1}, m_{t_2},
\cdots, m_{t_k})$. The time limitation mask $M_T$ masks the mission when the flight time of
returning to the depot after completing the mission exceeds the remained flight time of the
UAV, expressed as:

$$m_{t_j} = \begin{cases} 1 & \text{if} \quad T_j + T_{Return,j} > T_{Remain}, \\ 0 & \text{otherwise} \end{cases}, \qquad (j = 1, \ldots, k) \qquad (5)$$

where $T_j$ is the time to complete the mission $j$ from the current mission, $T_{Return,j}$ is the flight
time of returning to the depot time after completing the current mission, and $T_{Remain}$ is the
remained flight time of the UAV. The masking strategy generates the mask $M = M_C | M_T$,
where the operator | means the element-wise logical 'or' operation. Figure 5 shows an
example of the masking strategy. Each circle in the figure represents an arbitrary mission.
The completion mask $M_C$ and the time limitation mask $M_T$ are represented as red circles
and purple circles, respectively. The agent in the example only can select unmasked
missions to complete or the depot to return.

If every element of $M_T$ is masked before finishing the whole mission, the UAV is forced to return to the depot for refueling/recharging itself. When the UAV arrived at the depot, the remained flight time of the UAV is initialized, and then every element of $M_T$ is calculated by (5). After that, the UAV continues the subsequent tasks. Note that time for recharging is not explicitly considered in the cost; we assumed that the recharging can be done quickly by replacing the battery with a new one. However, if needed, we could easily include the recharging cost in the optimization problem formulation.

## Model architecture

The neural network model $\pi_\theta$ which approximates the VRP solver is parameterized with $\theta$. The policy with the model can be represented as the probability $p_{\pi_\theta}(\Omega|s)$, where $s = (v_1, v_2, \cdots, v_k)$ is the given mission nodes as the input of the model and $\Omega = (n_1, n_2, \cdots, n_k)$ is the output of the model, which is the permutation of the index of $s$. With the chain rule, the probability can be factorized as:

$$p_{\pi_\theta}(\Omega|s) = \prod_{t=1}^{k} p_{\pi_\theta}(\Omega_t|s, \Omega_{1:t-1}), \qquad (6)$$

where $\Omega_t$ is the output value at $t \in \{1, \cdots, k\}$ and $\Omega_{1:t-1}$ is the partial sequence of $\Omega$.

We utilize the Transformer style model architecture of *Kool, van Hoof & Welling (2019)* to approximate Eq. (6). The model takes the input which is a set of mission node data for the encoder and outputs the solution sequence with the decoder while satisfying the constraints. The encoder is implemented with multi-head attention (MHA) layers (*Vaswani et al., 2017*), and it generates the embeddings of each input element $(h_{e1}, h_{e2}, \cdots, h_{ek})$ where the embeddings represents the relationship among all of the other elements in the input mission nodes. To generates the embeddings with the encoder, the MHA layer utilizes query $Q_i = w_q v_i$, key $K_i = w_\kappa v_i$, and value $V_i = w_v v_i$ vectors, where $w_q$, $w_\kappa$, and $w_v$ are linear layers for projecting mission node features. The attention mechanism inferences the relationship between query and key by calculating the attention score as:

$$u_{ij} = \frac{Q_i^T \cdot K_j}{\sqrt{d}}, \qquad (j = 1, \cdots, k) \qquad (7)$$

where $u_{ij}$ is the attention score and $d$ is the embedding size. The attention score represents the similarity between $Q_i$ and $K_i$. Using the attention score, the embedding $h_{ei}$ of $v_i$ and the context vector $h_c$ are calculated as:

$$a_{ij} = softmax(u_{ij}), \qquad (8)$$

$$h_{ei} = \sum_{j=1}^{k} a_{ij} V_j, \qquad (9)$$

$$h_c = \frac{1}{k} \sum_{n=1}^{k} h_{en}. \qquad (10)$$

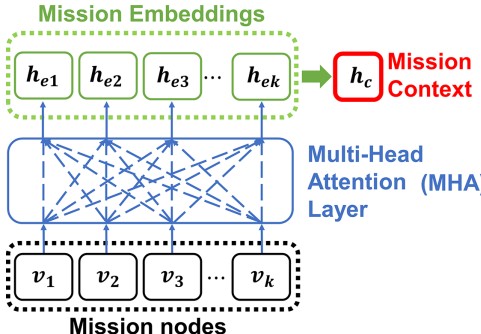

**Figure 6** **The encoder embeds the input mission nodes into embeddings with MHA.**

Note that the calculation of the attention mechanism is parallelized by the heads which are part of the MHA layer. In this study, the encoder consists of 3 MHA layers with eight heads and 128 embedding sizes. Figure 6 shows the embedding process of the encoder.

At each decoding step $t$, the decoder embeds the outputs from the encoder into $(h'_{e1}, h'_{e2}, \cdots, h'_{ek})$ with a MHA layer. Then, decoder selects the next node of the solution with the attention mechanism as described in *Vinyals, Fortunato & Jaitly (2015)*. In this case, the query vector $Q'$ consists of the context vector $h_c$, the partial solution information $\Omega_{1:t}$ which is abstracted as $\Omega' = (h_{en_1}, h_{en_t})$, where $n_t \in \Omega_{1:t}$, and the remained flight time budget $T_{Remain}$ for considering the flight time budget constraint. Note that $\Omega'$ is initialized as $(0, 0)$ before selecting the first mission to complete and then updated as $(h_{en_1}, h_{en_t})$, where $h_{en_1}$ is the embedding of the first solution node and $h_{en_t}$ is the embedding of the last solution node. This is because the agent of the VRP only needs to consider the uncompleted missions with respect to the last completed mission regardless of completed missions (*Kool, van Hoof & Welling, 2019*). Then, the probability of selecting each mission node is obtained by the attention score between the embeddings $(h'_{e1}, h'_{e2}, \cdots, h'_{ek})$, $Q'$, and mask $M$ from the masking strategy as:

$$u'_i = \begin{cases} -\infty & \text{if} \quad m_i = 1 \\ \dfrac{Q^T \cdot h'_{ei}}{\sqrt{d'}} & \text{otherwise} \end{cases}, \qquad (11)$$

$$a' = softmax(u'), \qquad (12)$$

where $u'_i$, $d'$ and $a'$ are the attention score, embedding size of the decoder, and the probability of selecting each mission, respectively. The next solution $n_t$ is selected by sampling from the probability distribution $a'$ and added to the last index of $\Omega_{1:t-1}$ to construct $\Omega_{1:t}$. After selecting the next solution, $T_{Remain}$ is reduced by the completion time of the selected mission and $\Omega'$ is updated with $\Omega_{1:t}$. In this study, the decoder consists of 1 MHA layer with one head and 128 embedding size. Figure 7 shows the example of decoding steps.

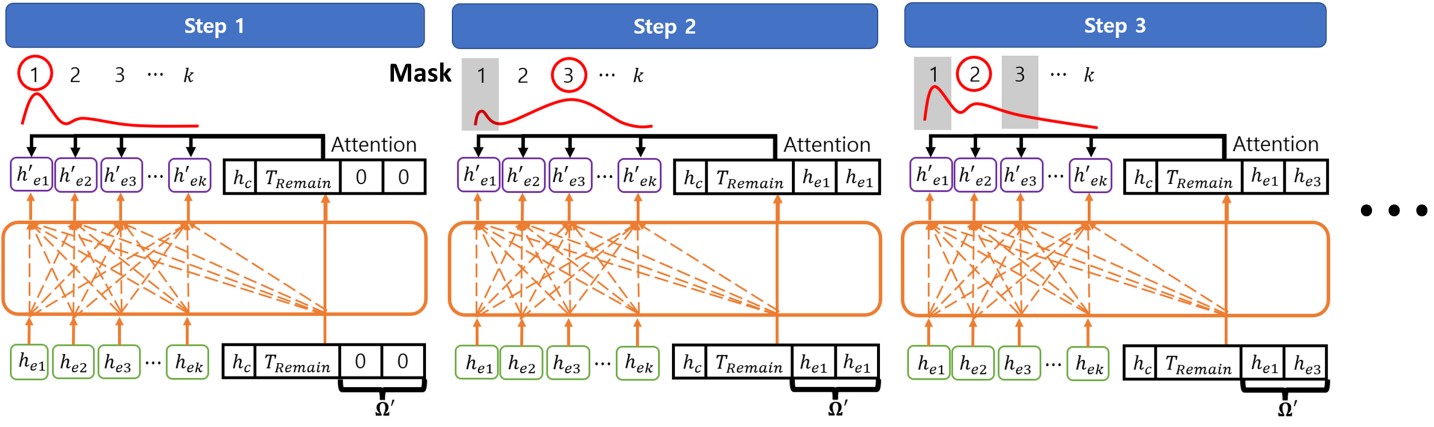

**Figure 7 The example of decoding steps.**

## REINFORCE with baseline

To update the neural network model, we use the REINFORCE algorithm (*Sutton et al., 2000*). In the Markov decision process (MDP) tuple $< s, a, r, \tau, \pi >$ for reinforcement learning, the state $s$ is the mission state, the action $a$ is the selected mission from the agent policy $\pi_\theta(a|s)$ which is the neural network model parameterized with $\theta$, the reward $r$ is the cost in Eq. (1), and the transition probability $\tau = p(s'|s, a)$ is the next state after selecting $a$ given $s$. Note that $\tau$ is deterministic in this work.

Since the REINFORCE algorithm produces the high variance gradient such that the algorithm might converge extremely slow during training, the baseline $b$ is utilized to reduce the variance. Paremeters $\theta$ are updated with the policy gradient method as:

$$\nabla_\theta J(\theta) \approx \frac{1}{N} \sum_{i=1}^{N} \sum_{t=1}^{k} \nabla_\theta \log \pi_\theta \left( \mathbf{a}_{i,t} | \mathbf{s}_{i,t} \right) \left( \sum_{t'=t}^{k} r_{i,t'} - b_i \right), \tag{13}$$

$$\theta \leftarrow \theta + \alpha \nabla_\theta J(\theta), \tag{14}$$

where $N$ is the number of batch size, $k$ is the number of mission, $\alpha$ is the learning rate, and $b$ is the baseline. Note that the baseline $b$ of this study is moving average of the cost during training (*Kool, van Hoof & Welling, 2019*). The proposed algorithm is trained with 1,280,000 mission instances with 512 batch size, 100 epochs, 1e−4 learning rate with the Adam optimizer (*Kingma & Ba, 2015*).

## NUMERICAL SIMULATIONS

This section provides the comprehensive simulation results to show the performance of the proposed approach. Every simulation is run on NVIDIA GeForce RTX 2080 GPU, Intel(R) i9-9900KF CPU, and 64 GB RAM.

The neural netowork is trained with the different number of missions in the range of (3, 30). The position of every mission is generated randomly in (0, 1) scaled two-dimensional (2-D) map with a uniform distribution. The coverage mission's radius is generated randomly in the range of (0.04, 0.08) with the uniform distribution. The place-position of

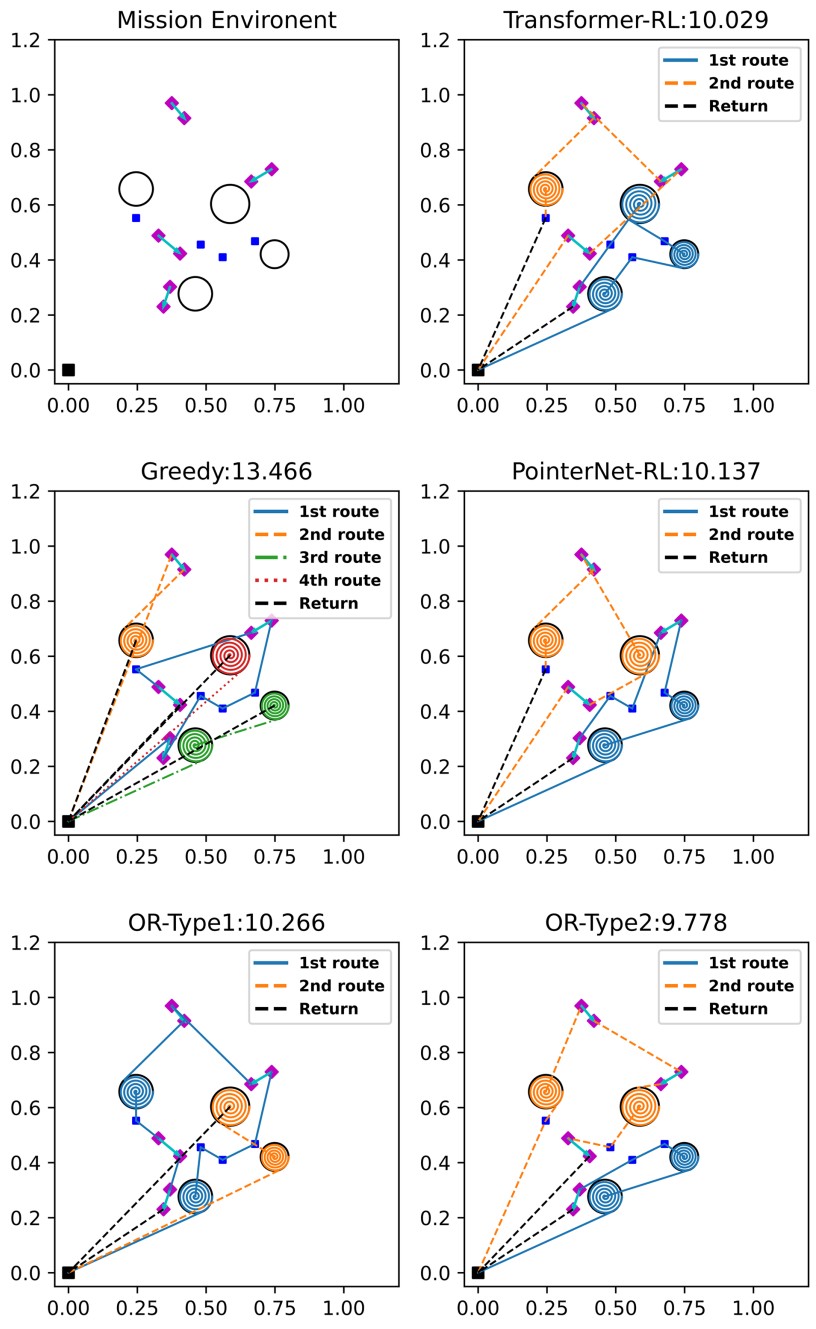

**Figure 8 Sample solutions of the different algorithms on the mission environment for 12 missions.** (four visiting, four coverage and four delivery).

the delivery mission is distant from the pick-position in the range of $(-0.1, 0.1)$ with the uniform distribution. The position of the depot is the origin without loss of generality. The velocity of the UAV is 1 and the flight time budget is 6.

We compare our algorithm (termed as Transformer-RL) with the Google OR-Tools (https://developers.google.com/optimization/routing/vrp) which is the state-of-the-art solver for the combinatorial optimization problem. We modified the software into two

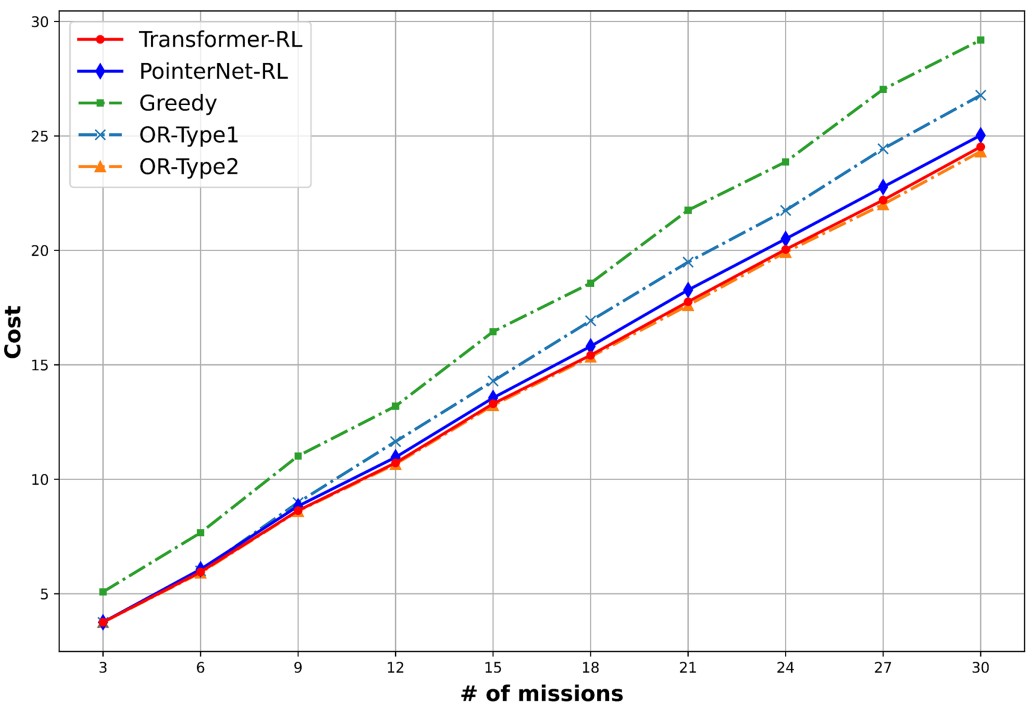

**Figure 9 Mean cost performance of each algorithm.**

types. The first baseline algorithm (OR-Type1) solves the given mission as a distance-limited VRP with a single-vehicle. The OR-Type1 generates a single route per iteration within the flight time budget. We give the penalty cost to the uncompleted missions to prevent generating an empty route. The penalty makes the algorithm try to generate a shorter route while satisfying the flight time limitation. Then, the OR-Type1 makes a plan iteratively until every mission is completed. The second baseline algorithm (OR-Type2) solves the mission instance as a distance-limited VRP setting, assuming that the number of available vehicles and the number of missions to be performed are the same. The assumption reduces the effort to solve the problem iteratively unlike the OR-Type1. Thus, the OR-Type2 generates multiple routes at once that complete every mission while deciding the desirable number of vehicles to utilize. We also compared ours with the simple greedy algorithm and the Pointer network-based reinforcement learning algorithm (PointerNet-RL) (*Bello et al., 2017*). The simple greedy algorithm selects the next mission with the lowest cost from the current mission node while satisfying the flight time limitation, and the PointerNet-RL uses RNN (*Vinyals, Fortunato & Jaitly, 2015*) for the neural network structure instead of the attention mechanism used in this study. Figure 8 visualizes the sample solution and total cost of each algorithm. Note that the return path to the depot of each route is represented with the black dashed line. In Fig. 8, the OR-Type2 generates the best solution which has the lowest cost, and reinforcement learning-based algorithms show better performance than that of the OR-Type1 and the greedy algorithm. The OR-Type1 generates each route while completing the most missions possible and the

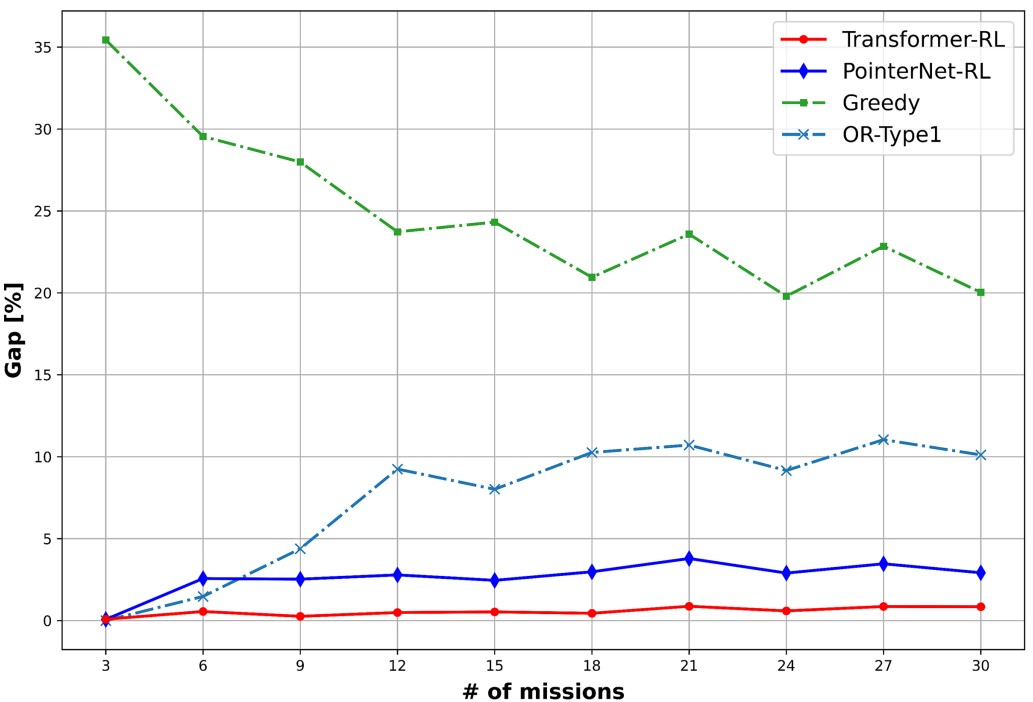

**Figure 10  Cost gap (%) of the algorithms with respect to the OR-Type2.**

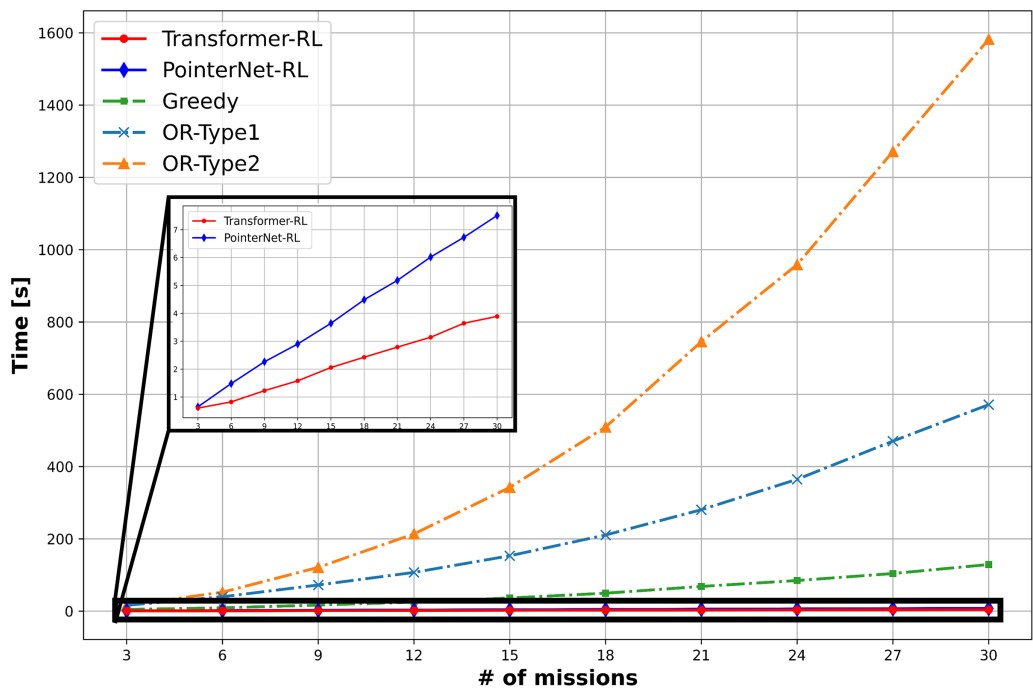

**Figure 11  Computation time.**

**Table 1 The performance of our proposed model compared with other algorithms.** The cos gap (%) is with respect to the OR-Type2.

| Algorithm | k = 9 | | | k = 15 | | | k = 21 | | |
|---|---|---|---|---|---|---|---|---|---|
| | Cost | Gap (%) | Time (s) | Cost | Gap (%) | Time (s) | Cost | Gap (%) | Time (s) |
| Transformer-RL | 8.76 | 1.846 | 1.22 | 13.55 | 2.461 | 2.05 | 18.14 | 3.08 | 2.78 |
| PointerNet-RL | 8.82 | 2.525 | 2.26 | 13.55 | 2.453 | 3.64 | 18.26 | 3.78 | 5.18 |
| Greedy | 11.01 | 27.987 | 18.62 | 16.44 | 24.314 | 41.67 | 21.75 | 23.58 | 85.61 |
| OR-Type1 | 8.98 | 4.381 | 85.23 | 14.29 | 8.012 | 173.73 | 19.48 | 10.71 | 315.71 |
| OR-Type2 | 8.60 | 0.00 | 144.36 | 13.23 | 0.00 | 384.49 | 17.60 | 0.00 | 789.88 |

greedy algorithm makes the most number of routes, which is inefficient due to its myopic strategy.

The total cost of the solution defined in Eq. (1) and the computation time is used to measure the performance of the algorithms. We use 10,000 mission instance samples to test the performance with the different number of missions in the range of (3, 30). Figure 9 provides the result of the performance analysis. The OR-Type2 shows the best performance in terms of the total cost, while Transformer-RL has a similar performance with the OR-Type2. Figure 10 shows Transformer-RL is better than the PointerNet-RL more clearly by the cost gap analysis with respect to the OR-Type2. Figure 11 provides the computation time for each algorithm. The computation time of the OR-Type2, which is the best algorithm for the cost, grows exponentially along with the scale of the mission. On the other hand, the Transformer-RL and the PointerNet-RL show significantly faster computation time than that of the other algorithms. The greedy algorithm also shows fast computation time, but it has the worst cost performance. Table 1 summarizes statistical results for a certain number of missions with the cost performance, the performance gap, and the computation time.

## CONCLUSIONS AND FUTURE WORK

In this article, we proposed an algorithm for mission planning of heterogeneous missions for a single UAV. We formulate the mission planning problem into a vehicle routing problem that has various methods to solve. We used an attention-based deep reinforcement learning approach, expecting fast computation time and sufficiently good performance. The numerical experiments show that the proposed algorithm can be a good selection with the reasonable trade-off between performance and computation time. However, as the proposed algorithm considers a deterministic mission environment and deals with a single UAV, our future work will consider the uncertainty of the mission environment such as the effect of the weather conditions and the operation of multiple UAVs with multi-agent reinforcement learning approaches.

### Funding

This research was supported by the Basic Science Research Program through the National Research Foundation of Korea (NRF) funded by the Ministry of Education (2020R1A6A1A03040570) and the Unmanned Vehicles Core Technology Research and Development Program through the National Research Foundation of Korea (NRF), Unmanned Vehicle Advanced Research Center (UVARC) funded by the Ministry of Science and ICT, the Republic of Korea (2020M3C1C1A01082375). The funders had no role in study design, data collection and analysis, decision to publish, or preparation of the manuscript.

### Grant Disclosures

The following grant information was disclosed by the authors:
Ministry of Education: 2020R1A6A1A03040570.
National Research Foundation of Korea (NRF).
Ministry of Science and ICT.
Republic of Korea: 2020M3C1C1A01082375.

### Competing Interests

Hyondong Oh is an Academic Editor for PeerJ.

### Author Contributions

- Minjae Jung conceived and designed the experiments, performed the experiments, analyzed the data, performed the computation work, prepared figures and/or tables, authored or reviewed drafts of the article, and approved the final draft.
- Hyondong Oh analyzed the data, authored or reviewed drafts of the article, and approved the final draft.

### Data Availability

The data is available at GitHub:
https://github.com/keep9oing/Heterogeneous-Mission-Planning.

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
