# Peer review of "Heterogeneous mission planning for a single unmanned aerial vehicle (UAV) with attention-based deep reinforcement learning"

_PeerJ Computer Science, doi:10.7717/peerj-cs.1119_

## Round 0.1 · original submission · Major Revisions

Dear authors, please address the comments of reviewers carefully. In addition, the authors may consider the following points,
1. Elaborate on the contribution of your work.
2. Proofread your manuscript carefully.
3. Consider the latest and related references.

Reviewer 1 ·

Basic reporting

no comment

Experimental design

no comment

Validity of the findings

no comment

Additional comments

1、 The English language should be improved to ensure that an international audience can clearly understand your text. Some examples where the language could be improved include lines 101, 132-133 – the current phrasing makes comprehension difficult.
2、 Your introduction needs more detail. I suggest that you improve the description at lines 35- 36 to provide more justification for your study. Mission planning problem is usually a kind of complex combinatorial optimization problem. It is difficult to describe these problems as VRP problems.
3、 Is the heterogeneous mission planning studied in this paper aimed at a single UAV? How to embody multiple UAVs, as the author wrote in the title? If it is only for a single UAV, I think it is no different from the traditional VRP Problem.
4、 How does the algorithm represent the process of the UAV returning to the base for charging and then continuing the subsequent tasks? This is not well reflected in the masking strategy designed by the author.
5、 I think the author has provided a good heterogeneous mission scenario and an effective reinforcement learning algorithm to solve this problem, but the author should make a more detailed design and improvement in the number of UAVs involved and how to reflect the charging strategy of UAVs returning to the base.

Reviewer 2 ·

Basic reporting

The article "Heterogeneous mission planning of UAVs with attention-based deep reinforcement learning" is contributing to the body of knowledge, the idea is clear. However, there is less literature review, needs to add more literature. Needs proper proofreading as there are grammatical mistakes.

Experimental design

The experimental designs are up to the mark

Validity of the findings

Proper literature should be studied and comparison should be carried out. As discussed earlier, the literature support is less in this research work.

Additional comments

The overall paper's idea is good and contributes to the body of knowledge, however, more literature should be added, and the result and findings should be supported by the literature. Review and proofread the paper thoroughly before final submission.

---

## Round 0.2 · accepted · Accept

Dear authors, we would congratulate to you on the acceptance of your research work. It is recommended to consider thorough proofreading with your final copy.

Reviewer 1 ·

Basic reporting

no comment

Experimental design

no comment

Validity of the findings

no comment

Additional comments

I think the author has carefully revised the paper based on the reviewer's comments. The overall paper's idea is good and contributes to the body of knowledge, the logical structure is reasonable, which conforms to the good requirements of academic paper writing. Therefore, I recommend acceptance of this paper.

Reviewer 2 ·

Basic reporting

The work has been improved

Experimental design

The comments are addressed and have been improved

Validity of the findings

Up to the mark

Additional comments

No comments